# Studies on the Relationships between Growth and Gonad Development during First Sexual Maturation of *Macrobrachium nipponense* and Associated SNPs Screening

**DOI:** 10.3390/ijms25137071

**Published:** 2024-06-27

**Authors:** Sufei Jiang, Yinxiang Xie, Zijian Gao, Yunpeng Niu, Cheng Ma, Wenyi Zhang, Yiwei Xiong, Hui Qiao, Hongtuo Fu

**Affiliations:** 1Key Laboratory of Freshwater Fisheries and Germplasm Resources Utilization, Ministry of Agriculture and Rural Affairs, Freshwater Fisheries Research Center, Chinese Academy of Fishery Sciences, Wuxi 214081, China; jiangsf@ffrc.cn (S.J.); zhangwy@ffrc.cn (W.Z.); xiongyw@ffrc.cn (Y.X.); 2Wuxi Fisheries College, Nanjing Agricultural University, Wuxi 214081, China; 18851976889@163.com (Y.X.); gaozijiangenomics@163.com (Z.G.); niuyp5259@163.com (Y.N.); 18225489231@163.com (C.M.)

**Keywords:** *Macrobrachium nipponense*, single-nucleotide polymorphism, Cathepsin L1, growth, sexual maturation

## Abstract

In this study, we used full-sib families to investigate the association between growth and gonad development during first sexual maturation of *M. nipponense*. We found that male GSI was significantly negatively correlated with growth traits (*p* < 0.01) and there were no significant correlations between female GSI (Gonadosomatic index) and growth traits (*p* > 0.05). HSI (Hepatopancreas index) in both males and females showed no significant correlations with growth traits (*p* > 0.05). We furthermore investigated the association between the specific allele of *Mn-CTS L1* polymorphism and gonad development and growth traits. In total, 35 mutation loci were screened and 16 high-quality single-nucleotide polymorphisms (SNPs) loci were obtained after validation. Four and two SNPs proved to be strongly associated with all growth traits in female and male *M. nipponense* separately, among which A+118T might be a candidate SNP positively associated with large growth traits. Two and one SNPs were screened, respectively, in males and females to associate with GSI, while three SNPs were detected to associate with female HSI, among which A+1379C may be applied as a potential molecular marker for gene-assisted selection to improve both reproduction speed and growth traits in *M. nipponense*.

## 1. Introduction

The oriental river prawn *Macrobrachium nipponense* is widely distributed throughout China [1]. It is an important economical freshwater species and is cultivated due to its strong adaptability and disease resistance and high economic value [2,3]. *M. nipponense* has the characteristic of rapid sexual maturation. The breeding season of *M. nipponense* is from April to October every year (with water temperature at 22 to 30 °C). The overwintering adult female prawns (BW ± SD: 0.78 ± 0.23 g, BL ± SD: 28.58 ± 3.35 cm) enter into the breeding season at the end of April, and its sexual maturation and embryo development will be accelerate by the rise in the water temperature. Every year, the newborn prawns from July in the pond can reproduce a large number of offspring in autumn (August to September, water temperature at 28 to 30 °C, BW ± SD: 0.50 ± 0.15 g, BL ± SD: 25.90 ± 2.12 cm) (commonly known as “autumn reproduction”). During the “autumn reproduction”, hatching to sexual maturation can take as little as 45 days, and rapidly maturing female prawns, with a body length as short as 3 cm, are able to lay eggs. This can result in multiple generations in a single pond, increasing food coefficient and hypoxia risk and reducing the ability of *M. nipponense* to tolerate adverse conditions [4]. “Autumn reproduction” can also aggravate cannibalistic behaviors. Such effects reduce the overall survival rate and the proportion of larger sized prawns. Therefore, research has focused on developing new strains of *M. nipponense* characterized by fast growth but late maturation to avoid these negative impacts on *M. nipponense* aquaculture.

Gonad development is the basis for the reproductive behavior of fishery resources, and it is closely related to the body size and age of individuals. In fish, most studies showed that first sexual maturation is related to body size (body length). Primary maturation begins when the size of the fish reaches or exceeds the minimum body length required for sexual maturity, and then the proportion of mature individual increases with the increase in body length. There is some research which indicated that first sexual maturity of fish, such as *Maccullochella peeli peeli* and *Larimichthys crocea*, were closely related to age [5,6,7]. In *L. crocea*, it has been proven to have a negative correlation with body weight, and selective breeding for the specific gonadosomatic index (GSI) is beneficial in improving the dressing percentage [6,7]. In crustaceans, there are a few reports about the relationship between first sexual maturity and body size and age, confirming that body size is more vital than age in *Penaeus schmitti*, *Oratosquilla oratoria* and *Trachysalambria curvirostris* [8,9,10]. In *T. curvirostris*, the growth of individuals was slowed but did not stop completely during sexual maturity [10]. In *M. nipponense*, GSI had a significant negative effect on the size of female prawns [11]. Therefore, elucidating the correlation between gonadal development and growth of fishery resources could provide a scientific basis for resource assessment and breeding management.

Crustacean gonad development occurs in two stages. The first relates to the early stage of first sexual maturity, which synchronizes with growth and involves complex changes in behavior, external morphology and gonadal morphology; by contrast, the second relates to periodic sexual maturity following adult maturation [12]. Various factors, such as temperature, dietary and salinity, involved in the growth of *M. nipponense* have been reported [13,14,15,16,17]. In addition, research has focused on the genetic mechanisms of gonad development in *M. nipponense* within numerous genes, such as those encoding vitellogenin (Vg), cathepsin (Mn-CTS L1) and legumain-like protease (Lel), being found to be closely related to rapid sexual maturation in this species [4,18,19]. However, less research has focused on the relationship between growth and reproduction in *M. nipponense*. Preliminary association analyses between gonad development and growth traits in randomly sampled *M. nipponense* populations identified several SNPs based on Vg and associated with ovarian development and growth [19]. Although these results provide a preliminary understanding of the relationship between the gonadal development and growth of *M. nipponense*, how the variation between reproduction and growth is regulated was not fully elucidated.

In this study, we performed a correlation analysis between growth and gonad development during first sexual maturation of *M. nipponense*, and genetic effects studies of Cathepsin L gene on growth and gonad development were also carried out to find the SNPs for march-assisted selection. The results of this study will provide additional scientific background information for basic biological research of *M. nipponense*. These results could also accelerate selection of novel varieties of *M. nipponense* by providing effective tools for genetic improvement.

## 2. Results

### 2.1. Growth and Gonad Development Trait Analyses

In total, trait data from 392 *M. nipponense* were obtained (195 males and 197 females) (Table 1). The results showed that all eight growth traits were significantly higher in males than in females during first sexual maturation (*p* < 0.01). By contrast, the GSI showed the opposite result, being significantly higher in females than in males (*p* < 0.05). There were no significant differences between the sexes in terms of HSI results (*p* > 0.05). Q-Q plots showed normal distribution of growth and gonad development traits in males and females (Appendix A).

### 2.2. Association between Growth and Gonad Development Traits

Correlation analyses were performed using GSI and HSI (Table 2). In males, there were significant negative correlations between GSI and all eight growth traits (*p* < 0.01), whereas there were no significant correlations between HSI and all eight growth traits (*p* > 0.05). In females, there were no significant correlations between GSI and all eight growth traits or between HSI and all eight growth traits (*p* > 0.05).

### 2.3. SNP Identification and Polymorphism of Mn-CTS L1

Based on sequence alignment and PCR verification, the complete DNA sequence of Mn-CTS L1 was 2013 base pairs (bp) in length and contained six exons (960 bp in total) and five introns (1053 bp in total). Following the detection of SNP sites, 35 mutation loci were screened. In total, 16 high-quality SNP loci were obtained after removing loci with <10% mutation frequency, based on PCR validation. The length of each exon and intron, GC content and distributions of these 16 SNPs are detailed in Table 3. A/G and C/T transitions were 25.0% and 25.0%, respectively, whereas the transversions of A/T, A/C, C/G and G/T were 18.75%, 12.50%, 12.50% and 6.25%, respectively.

The genotypes and their frequencies (Appendix A) were calculated based on direct sequencing in the separate male and female populations. The results indicated that genotype frequencies did not differ between males and females, with AG in A+782G, GC in C+1698G and AG in A+1885G being rare genotypes in female *M. nipponense*.

Polymorphism analyses of the 16 SNP loci in male and female populations separately were performed using Popgene32 (Appendix A). In males and females, Ne ranged from 1.3161 to 1.9925 and from 1.3314 to 2.0070, respectively; Ho ranged from 0.5006 to 0.7592 and from 0.4970 to 0.7505, respectively; and He ranged from 0.4219 to 0.7774 and from 0.4008 to 0.7728, respectively. PIC revealed that most of the 16 SNPs were moderately polymorphic, with a few that were minimally polymorphic.

### 2.4. Association between Growth and Gonad Development Traits

TL, BW, BL, SPL, AW, AL, CW and CL of 195 males and 197 female *M. nipponense* were measured, and an association analysis was performed to assess the relationship between these growth traits and Mn-CTS L1 SNPs. In females, 11 SNPs were significantly associated with growth traits (*p* < 0.05), including A+118T, C+155G, T+216G and A+1379C (Table 4). Of these loci, all loci except T+216G were significantly associated with all eight growth traits. T+216G was only significantly associated with BL, SPL and CL (*p* < 0.05). In A+118T, all growth traits of the AT and AA genotypes were significantly higher compared with the TT genotype (*p* < 0.05), and the growth traits of the AT genotype were higher compared with the other genotypes in the remaining loci. GG and GC genotypes in C+155G and the CC genotype in A+1379C were positively associated with eight growth traits (*p* < 0.05). In males, there were two SNPs significantly associated with growth traits (*p* < 0.05): A+118T and A+1379C (Table 4). All loci were significantly associated with all eight growth traits. In A+118T, all growth traits of the AT genotype were significantly higher compared with the AA and TT genotypes (*p* < 0.05) and were higher compared with the other genotypes at the remaining loci. The CC genotype in A+1379C was positively associated with all eight growth traits (*p* < 0.05).

### 2.5. Association of Mn-CTS L1 SNPs with Gonad Development Traits

GSI and HSI of 195 males and 197 female *M. nipponense* were measured and association analysis was performed to assess the relationship between these gonadal development traits and *Mn-CTS L1* SNPs (Table 5). In females, two SNPs were significantly associated with GSI (A+782G and A+1379C). The GSI of the AA genotype in A+782G was significantly lower compared with that of the AG genotype (*p* < 0.05), whereas that of the CC and AC genotypes in A+1379C was significantly lower compared with that of the AA genotype (*p* < 0.05). In addition, these two SNPs were also significantly associated with HSI. The HSI of the AA genotype in A+782G was significantly higher compared with that of the AG genotype (*p* < 0.05). The HSI of the AC genotype in A+1379C was significantly lower compared with that of the AA genotype (*p* < 0.05), whereas that of the AG genotype in A+206G was significantly higher than that of the GG and AA genotypes (*p* < 0.05). In males, only A+1379C was significantly positively associated with GSI in the AA genotype (*p* < 0.05). No SNP was found to be significantly associated with FSI in either sex (*p* > 0.05).

## 3. Discussion

Sexual maturation is an important transition from juvenile developmental growth to adult reproductive growth [20]. The gonadosomatic index (GSI) is an economically important trait in many aquatic animals, which is related to the size of fishery resources and their sexual maturity. Growth trait parameters at sexual maturity are one of the key parameters for fishery resource assessment and risk assessment of management strategies [21]. They are widely used in the analysis of growth pattern changes, estimation of maximum BL, and determination of minimum opening specifications of fishery resources, which provide an important reference for sustainable utilization of fishery resources [22,23,24,25]. Research has suggested that growth traits are affected by sexual maturity (including first sexual maturity and adult periodic sexual maturity), resulting in increasing pressure on resource utilization [26]. A series of linear models and estimation methods have been established and are widely applied in fishery research [26,27,28].

However, the correlation of gonadal development and growth was not same in different species. In crustacean *Oratosquilla oratoria*, the gonad of large-size individuals developed significantly earlier than that of small-size individuals [7]. In contrast, in fish, such as the large yellow croaker, body weight has been reported to show a very strong negative correlation (−0.96) with GSI [9,10]. Hence, selective breeding for specific GSI is beneficial in improving dressing percentage. In previous work, we used a randomly sampled juvenile *M. nipponense* population to investigate the association between ovarian development and growth [28]. The results showed that female GSI was negatively correlated with growth traits, and there were no significant correlations between male GSI and growth. These results were in accordance with the miniaturization observation in females with fast gonad development during autumn reproduction. However, the development of this randomly selected *M. nipponense* population was non-synchronized, especially in the multigenerational population inhabiting the same aquaculture pond, making it difficult to fully assess the relationship between reproduction and growth. In the current study, full-sib families were used to investigate the association between growth and gonad development during first sexual maturation of *M. nipponense*. The results indicate no significant correlations between ovarian development and growth traits, whereas male growth traits are positively affected by sexual maturity during growth in the first sexual maturation stage of *M. nipponense*. The hepatopancreas, which is an important source of energy for gonadal development, showed no correlation with growth traits during first sexual maturation of *M. nipponense*. These results also proved that the individuals with fast gonad development do not have a tendency to miniaturize. All the reports indicated that the effects of gonadal development on growth are the opposite in vertebrates and invertebrates, and its mechanism needs further research.

Using SNPs in association studies is a common strategy to screen the major gene and quantitative trait loci that regulate polygenic traits [29,30]. Significant associations between gene polymorphisms and a specific allele provide strong evidence that the genes are involved in regulating the target traits [31]. Such an approach has been successfully applied in many aquacultural species of both fish and crustaceans [32,33,34,35,36,37]. Cathepsin Ls, which belongs to the lysosomal papain C1 family, is a cysteine protease with important roles in pathological and physiological processes, such as protein hydrolysis, antigen presentation, proteolysis and tumor metastasis [38,39]. Cathepsin L is the main enzyme involved in Vg degradation in insects [40,41] and has also been reported to be involved in Vg processing in zebrafish [42]. In our previous study, the gene encoding cathepsin L1 in *M. nipponense* (Mn-CTS L1) was characterized, and the RNAi results showed that it can promote ovarian maturation [17]. In the current study, we investigated the association between specific alleles of Mn-CTS L1 polymorphisms and economically important traits of *M. nipponense*.

The current results showed that four SNPs were significantly associated with all female growth traits, whereas two SNPs were significantly associated with male growth traits. In both the male and female populations, A+118T and A+1379C were strongly associated with all growth traits in *M. nipponense*. All three genotypes in A+118T showed much larger growth trait values compared with other loci in both male and female populations. Thus, A+118T might be a candidate SNP positively associated with larger growth traits. Further analysis showed that the number of SNPs significantly associated with gonadal development differed in males and females. In both male and female populations, A+1379C was strongly associated with all gonadal development traits. A+1379C showed multiple associated characteristics following a combined comparison of SNPs associated with growth and gonadal development traits. The CC genotype in A+1379C was positively associated with eight growth traits but negatively associated with GSI (*p* < 0.05). Thus, A+1379C could be applied as a potential molecular marker for gene-assisted selection to improve both reproduction speed and growth traits in *M. nipponense.*

Growth and reproductive performance are two of the most important biological indicators in crustaceans but are often studied in isolation. In this study, we focused preliminarily on the intrinsic genetic association between growth and reproduction by using full-sib families during the first sexual maturation stage of *M. nipponense*. We raised sib families in different cages in the same pond to reduce the impact of environmental and population differences on the results. However, this is only a preliminary study under ideal conditions. In practical aquaculture applications, environmental factors such as temperature, water quality and diet, which could significantly influence growth and reproductive traits, and expanding the genetic diversity and number of families studied will provide a more comprehensive understanding of the genetic factors at play. The complex polygenic nature of these traits can involve numerous genes and environmental interactions. These identified SNPs and their associations with desired traits are likely to be only a small part of a complex genetic network and should be validated in different populations and under varied aquaculture conditions to ensure their applicability in broader genetic improvement programs.

The studies about the impact of different temperatures, water quality and diets on growth and reproductive traits will be further strengthened, and the underlying genetic mechanisms will be investigated. Diverse genes and large-scale family samples will be gradually introduced to give a more comprehensive understanding of the role of genetic factors in growth and reproductive performance. All the findings will be validated in different populations and under varied aquaculture conditions to make them more generalizable and applicable. Moreover, detailed morphological, physiological or behavioral aspects should be included to provide a more complete understanding of the genetic variation involved in *M. nipponense* biology and ecology. Future studies could benefit from using AI technologies to integrate genetic data with environmental and morphological analyses, offering a holistic view of the factors influencing *M. nipponense* growth and reproduction [43,44]. Finally, the identification of potential SNPs for marker-assisted selection is only the first step in aquatic genetic improvement; more considerations, such as cost, ethical implications of genetic selection and the balance between improving specific traits versus the overall health and resilience of populations, should be considered before further practical applications.

## 4. Materials and Methods

### 4.1. Ethics Statement and Consent to Participate

No endangered or protected species were involved in this study, and all experimental protocols and methods were approved in July 2022 (Authorization NO. 202207014009) by the Freshwater Fisheries Research Center Animal Care and Use Ethics Committee (Wuxi, China).

### 4.2. Sample Prawns

Healthy adult *M. nipponense* [*n* = 50, body weight (BW) ± SD: 1.32 ± 0.17 g; BL ± SD: 3 0.24 ± 3.20 g, 10 month age] were selected from the Freshwater Fisheries Research Center Dapu Scientific Experimental Base (Wuxi, China). Six full-sib families were constructed randomly with well-developed prawns (female:male = 1:1) without any signs of disease, injury or deformity. These full-sib offspring individuals were hatched in August 2022 and raised in cages in one pond under the same environmental conditions (the water temperature was from 22 to 31 °C, the pH was from 6.7 to 7.5, the dissolved oxygen content was from 4.5 to 11.7 mg/L and the ammonia nitrogen content was from 0.25 to 0.81 mg/L) and fed an artificial feed (5% of BW) twice daily at 08:00 h and 18:00 h. The culturing cycle lasted 77 days from August to October in 2022 until the female prawns reached their first sexual maturation, identified by the presence of mature ovaries (i.e., the ovaries went green and filled the carapace). The specific classification of different ovarian stages was based on previously published criteria [45]. One of the six full-sib families with more than 500 offspring was randomly selected as a sample family, from which 250 male and female prawns were randomly sampled for further investigation.

### 4.3. Traits Measurement and DNA Extraction

Growth and gonad developmental traits were measured in both the male and female prawns. The growth traits index included the following: Body weight (BW): the samples were weighted by removing water. Total length (TL): from the rostral front end to end of the telson. Body length (BL): from the boundary of the post-eyestalk to the tip of the telson. Second pereiopod length (SPL): from the second pereiopod base to its end. Abdomen width (AW): the width of the third abdomen. Abdomen length (AL): from the first abdomen front to the sixth abdominal segment end. Carapace width (CW): the width of carapace at the fifth pereiopod. Carapace length (CL): from the tip of the rostral front to the central end of the back of the carapace. The gonadal development index included the hepatopancreas index (HIS; hepatopancreas weight/BW × 100%) and gonadosomatic index (GSI; ovary weight/BW × 100%). The muscle tissues were stored at –80 °C until use.

DNA extraction quality testing was performed according to previous research [28].

### 4.4. Primer Design and Synthesis of Cathepsin L Gene and PCR Amplification

The Mn-CTS L1 sequence (accession number MW684082) was obtained from previous research [17] and compared with the *M. nipponense* reference genome (NCBI Accession No.: ASM1510439v1, https://ftp.cngb.org/pub/CNSA/data2/CNP0001186/CNS0254395/CNA0014632/, accessed on 22 May 2022). Premier 5.0 was used to design Mn-CTS L1 primers for amplifying the genomic sequence. All Mn-CTS L1 segments were verified by PCR validation. The primers, which are listed in Table 6, were synthesized by Sangon (Shanghai, China). PCR amplification reactions contained 2 μL (50 ng/μL) DNA template, 1 μL (10 μmol/L) forward and reverse primers, 12.5 μL PCR mix and 8.5 μL ddH_2_O. The PCR cycling conditions were 94 °C for 4 min, followed by 30 cycles of 94 °C for 5 s, 60 °C for 30 s and 72 °C for 30 s, then 72 °C 10 min (Eppendorf, Wesseling, German). The PCR products were tested by a 1.2% agarose gel and then sent to Sangon Biotech (Shanghai, China) for sequencing. The sequencing results were compared by MEGA 11.0 software to screen SNP sites.

### 4.5. Statistical Analysis

All growth and gonad developmental traits were tested for normal distributions by using SPSS 17.0 (*p* > 0.05). Correlation analyses between growth traits and gonadal development indices were also performed by using SPSS 17.0.

DNAMAN 6.0 combined with Chromas 2.4 was used for sequence alignments and SNP screening. Popgene32 was used to calculate the expected heterozygosity (He), effective number of alleles (Ne), and observed heterozygosity (Ho) [46]. The polymorphism information content (PIC) value was also calculated [47].

Correlation analyses between SNP genotypes and growth and gonad development traits were assessed using General Linear Model (GLM)-l in SPSS 17.0. Duncan’s multiple-range test was applied to calculate significant differences (at *p* < 0.05).

## 5. Conclusions

In this study, full-sib families were used to investigate the association between growth and gonad development during first sexual maturation of *M. nipponense*. Male GSI was significantly negatively correlated with growth traits (*p* < 0.01), whereas there were no significant correlations between female GSI and growth traits (*p* > 0.05). HSI in both males and females showed no significant correlations with growth traits (*p* > 0.05). Four and two SNPs of Mn-CTS L1 were strongly associated with all growth traits in female and male *M. nipponense*, respectively, with A+118T revealed as a candidate SNP positively associated with large growth traits. Two and one SNPs associated with GSI in females and males, respectively, whereas three SNPs were found to associate with female HSI, with A+1379C revealed as a potential molecular marker for gene-assisted selection to improve both reproduction speed and growth traits in *M. nipponense*.

## Figures and Tables

**Table 1 ijms-25-07071-t001:** Statistics of growth and gonad development traits in male and female populations.

	Total Length (TL, mm)	Body Length (BL, mm)	Body Weight (BW, mm)	Second Pereiopod Length (SPL, mm)	Abdomen Length (AL, mm)	Abdomen Width (AW, mm)	Carapace Length (CL, mm)	Carapace Width (CW, mm)	GSI%	HSI%
F	40.43 ± 5.23	27.58 ± 3.54	0.72 ± 0.33	22.24 ± 4.20	18.34 ± 2.47	4.87 ± 0.79	10.54 ± 1.69	5.58 ± 0.93	1.21% ± 0.75%	1.51% ± 0.11%
M	46.51 ± 9.29	31.79 ± 6.30	1.22 ± 0.82	28.72 ± 9.56	21.08 ± 4.34	5.49 ± 1.13	12.51 ± 2.89	6.52 ± 1.70	1.01% ± 0.83%	1.30% ± 0.09%
*p*	<0.01	<0.01	<0.01	<0.01	<0.01	<0.01	<0.01	<0.01	<0.05	>0.05

**Table 2 ijms-25-07071-t002:** Correlation analysis between growth and gonad development traits in male and female prawns.

Correlation		Total Length (TL, mm)	Body Length (BL, mm)	Body Weight (BW, mm)	Second Pereiopod Length (SPL, mm)	Abdomen Length (AL, mm)	Abdomen Width (AW, mm)	Carapace Length (CL, mm)	Carapace Width (CW, mm)
MGSI	*PC*	−0.336 **	−0.426 **	−0.309 **	−0.402 **	−0.304 **	−0.322 **	−0.391 **	−0.340 **
	*p*	0.001	0.001	0.003	0.005	0.004	0.002	0.007	0.001
MHSI	*PC*	0.091	0.121	0.083	0.076	0.108	0.135	0.142	0.112
	*p*	0.207	0.091	0.251	0.291	0.134	0.061	0.058	0.118
	*n*	195	195	195	195	195	195	195	195
FGSI	*PC*	−0.111	−0.112	−0.119	−0.024	−0.046	−0.094	0.003	−0.105
	*p*	0.122	0.166	0.096	0.733	0.519	0.188	0.965	0.143
FHSI	*PC*	0.041	0.054	−0.006	0.022	0.093	0.079	0.120	0.026
	*p*	0.569	0.453	0.929	0.758	0.193	0.269	0.094	0.718
	*n*	197	197	197	197	197	197	197	197

Notes: MGSI, male GSI; FGSI, female GSI; FHSI, female HSI; MHSI, male HSI; PC, the Pearson coefficient; *p*, two tails test; *n*: sample number; ** means level of 0.01.

**Table 3 ijms-25-07071-t003:** The length of each exon and intron, GC content and distributions of 16 SNPs.

Location	Length/bp	GC%	SNPs
Exon 1	68	49	
Intron 1	371	29	A+118T, C+155G, 206G, T+216G, A+333G
Exon 2	154	50	T+521A
Intron 2	177	35	C+640T, C+642T, T+698C, A+709T
Exon 3	180	54	A+782G
Intron 3	147	42	
Exon 4	187	51	
Intron 4	144	32	A+1379C
Exon 5	166	55	
Intron 5	214	35	C+1698G, C+1717T
Exon 6	205	54	A+1884C, A+1885G

**Table 4 ijms-25-07071-t004:** Association of *Mn-CTS L1* SNPs with growth traits in female and male populations.

SNP	Genotype	Total Length (TL, mm)	Body Length (BL, mm)	Body Weight (BW, mm)	Second Pereiopod Length (SPL, mm)	Abdomen Length (AL, mm)	Abdomen Width (AW, mm)	Carapace Length (CL, mm)	Carapace Width (CW, mm)
**Female**									
A+118T	TT	39.77 ± 4.83 ^b^	27.17 ± 3.27 ^b^	0.67 ± 0.28 ^c^	21.79 ± 3.88 ^b^	18.01 ± 2.18 ^b^	4.80 ± 0.74 ^b^	10.41 ± 1.66 ^b^	5.46 ± 0.84 ^c^
	AA	42.17 ± 5.83 ^b^	28.67 ± 4.23 ^b^	0.82 ± 0.41 ^b^	23.06 ± 5.01 ^b^	19.12 ± 3.18 ^b^	5.06 ± 0.84 ^b^	10.82 ± 1.82 ^b^	5.90 ± 1.11 ^b^
	AT	45.01 ± 5.37 ^a^	30.42 ± 3.54 ^a^	1.06 ± 0.42 ^a^	25.47 ± 4.72 ^a^	20.39 ± 2.66 ^a^	5.40 ± 0.94 ^a^	11.60 ± 1.39 ^a^	6.31 ± 1.09 ^a^
C+155G	GG	40.55 ± 5.41 ^a^	27.68 ± 3.69 ^a^	0.72 ± 0.34	22.28 ± 4.32 ^a^	18.38 ± 2.54	4.89 ± 0.78	10.60 ± 1.72 ^a^	5.58 ± 0.95
	CC	37.35 ± 3.63 ^b^	25.40 ± 2.52 ^b^	0.54 ± 0.14 ^b^	19.20 ± 2.11 ^b^	17.10 ± 1.63	4.49 ± 0.75	9.58 ± 1.76 ^b^	5.10 ± 0.81 ^b^
	GC	41.77 ± 3.82 ^a^	28.36 ± 2.43 ^a^	0.78 ± 0.27 ^a^	23.59 ± 3.60 ^a^	18.78 ± 2.16	5.02 ± 0.80	10.78 ± 1.24 ^a^	5.85 ± 0.72 ^a^
T+216G	GG	40.84 ± 5.62	27.97 ± 3.81 ^a^	0.74 ± 0.35	22.53 ± 4.49 ^a^	18.56 ± 2.62	4.91 ± 0.81	10.68 ± 1.77 ^a^	5.62 ± 0.99
	TT	39.27 ± 4.16	26.35 ± 2.82 ^b^	0.63 ± 0.19	20.44 ± 3.19 ^b^	17.69 ± 1.89	4.82 ± 0.79	9.92 ± 1.59 ^b^	5.45 ± 0.89
	GT	40.01 ± 4.09	27.13 ± 2.67 ^a^	0.70 ± 0.30	22.33 ± 3.54 ^a^	17.98 ± 2.11	4.80 ± 0.74	10.53 ± 1.37 ^a^	5.53 ± 0.73
A+1379C	AA	38.10 ± 3.96 ^b^	25.91 ± 2.59 ^b^	0.56 ± 0.16 ^b^	20.73 ± 3.00 ^b^	17.33 ± 1.88 ^b^	4.55 ± 0.61 ^b^	9.87 ± 1.48 ^b^	5.14 ± 0.68 ^b^
	CC	41.62 ± 5.50 ^a^	28.38 ± 3.75 ^a^	0.80 ± 0.37 ^a^	22.89 ± 4.56 ^a^	18.84 ± 2.64 ^a^	5.06 ± 0.82 ^a^	10.87 ± 1.68 ^a^	5.80 ± 0.99 ^a^
	AC	40.16 ± 4.16 ^a^	27.79 ± 2.72 ^a^	0.68 ± 0.18 ^a^	21.09 ± 4.34 ^a^	18.15 ± 1.73 ^a^	4.59 ± 0.59 ^b^	10.61 ± 2.08 ^a^	5.49 ± 0.64 ^a^
**Male**									
A+118T	TT	45.91 ± 9.06 ^b^	31.39 ± 6.12 ^b^	1.17 ± 0.82 ^b^	28.50 ± 9.57 ^b^	20.80 ± 4.21 ^b^	5.39 ± 1.09 ^b^	12.36 ± 2.77 ^b^	6.40 ± 1.63 ^b^
	AA	49.35 ± 10.84 ^b^	34.06 ± 7.07 ^a^	1.50 ± 0.91 ^b^	30.71 ± 10.81 ^a^	22.44 ± 4.61 ^a^	5.92 ± 1.17 ^a^	13.05 ± 3.41 ^a^	7.19 ± 2.03 ^a^
	AT	54.44 ± 8.82 ^a^	35.02 ± 6.08 ^a^	1.91 ± 1.04 ^a^	35.49 ± 11.20 ^a^	24.08 ± 4.87 ^a^	6.34 ± 1.15 ^a^	15.23 ± 3.30 ^a^	7.83 ± 2.04 ^a^
A+1379C	AA	43.73 ± 8.84 ^b^	29.89 ± 5.77 ^b^	0.99 ± 0.67 ^b^	26.47 8.10 ^b^	19.92 ± 4.06 ^b^	5.23 ± 1.09 ^b^	11.82 ± 2.82 ^b^	6.02 ± 1.53 ^b^
	CC	49.05 ± 9.75 ^a^	33.61 ± 6.68 ^a^	1.46 ± 0.94 ^a^	31.16 ± 10.91 ^a^	22.16 ± 4.54 ^a^	5.73 ± 1.16 ^a^	13.20 ± 3.11 ^a^	6.98 ± 1.83 ^a^
	AC	44.12 ± 6.02 ^b^	29.77 ± 4.15 ^b^	0.92 ± 0.48 ^b^	26.21 ± 4.78 ^b^	19.48 ± 2.47 ^b^	5.15 ± 0.96 ^b^	11.90 ± 1.45 ^b^	6.01 ± 1.36 ^b^

Notes: Different letters indicate significant difference of different genotypes of one same SNP site (*p* < 0.05).

**Table 5 ijms-25-07071-t005:** Association of *Mn-CTS L1* SNPs with gonad development traits in male and female.

SNP	Genotype	FGSI%	SNP	Genotype	MGSI%	SNP	Genotype	FHSI%
A+782G	AA	1.09 ± 0.64 ^b^	A+1379C	AA	1.23 ± 1.19 ^a^	A+206G	GG	5.80 ± 1.27 ^b^
	AG	1.32 ± 0.79 ^a^		CC	0.95 ± 0.61 ^b^		AA	5.51 ± 1.30 ^b^
A+1379C	AA	1.30 ± 0.70 ^a^		AC	1.03 ± 1.15 ^b^		AG	6.68 ± 3.25 ^a^
	CC	1.05 ± 0.40 ^b^				A+782G	AA	5.99 ± 1.96 ^a^
	AC	1.07 ± 0.43 ^b^					AG	5.38 ± 1.30 ^b^
						A+1379C	AA	5.99 ± 1.90 ^a^
							CC	5.84 ± 1.33 ^a^
							AC	5.03 ± 1.14 ^b^

Notes: MGSI, male GSI; FGSI, female GSI; FHSI, female HSI; MHSI, male HSI; Different letters indicate significant difference of different genotypes of one same SNP site (*p* < 0.05).

**Table 6 ijms-25-07071-t006:** Specific primers used to verify the SNP loci in the *Mn-CTS L1* gene.

Primer Name	Forward	Reverse	Amplification Location (bp)
CTS L1-1	TCTCCCGTTCAATAAACAT	TAAGTACCCTACCTGAACCACCT	1–336
CTS L1-2	TAAGTACCCTACCTGAACCACCT	TCAACATCGGCAGCTCTGG	285–994
CTS L1-3	TCCCTCCCACCATTCGTTT	TTGGCAGACTTCCGGTTTT	786–1213
CTS L1-4	AAGCAATGCCTCCCTGACA	GGACCTTCGTCGTGGACAG	1070–1607
CTS L1-5	TCTGCTGTCCACGACGAAG	ACACCGCAGTGGTTGTTCT	1585–2013

## Data Availability

Data will be provided upon request.

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
