# Peer review of "Studies on the Relationships between Growth and Gonad Development during First Sexual Maturation of Macrobrachium nipponense and Associated SNPs Screening"

_ijms, 2024, doi:10.3390/ijms25137071_

Round 1
Reviewer 1 Report
Comments and Suggestions for Authors
Review - ijms-3048148
The publication titled: „Studies on the relationships between growth and gonad development during the first sexual maturation of Macrobrachium nipponense and associated SNPs screening” by Sufei Jiang et al. is a very good study with application molecular analyses in aquaculture organisms. In my opinion minor revisions are needed for the paper’s acceptance. I have listed some suggestions that should be included in the text of the manuscript.
1. Line 36 - please explain if there are different times/seasons of breeding prawns. If so, please include it in the text. Please also include information on the seasonality of reproduction, including the length and body weight of organisms, and indicating the water temperature.
2. Paragraph in lines: 44-54 - Please describe clear differences and similarities between crustaceans and fish in terms of reaching sexual maturity at a given age in relation to body size. Moreover, the sentence (lines: 48-48) is unclear. Please explain the height of reproductively mature individuals.
3. Paragraph in lines: 44-54 – please add the fragment of text about GSI in crustacean. This is very important information to refer to the data regarding the large yellow croaker.
4. Lines 58-60 – Please add some explanations of various mechanisms involved in growth of M. nipponense.
5. Line 263 – Please add the information about age and length of selected M. nipponense.
6. Paragraph 4.3. - The information about the method of measurements of: total length (TL), BW, body length (BL), second pereiopod length (SPL), abdomen width (AW), abdomen length (AL), carapace width should be included (CW), and carapace length (CL) in this paper.
7. Lines 266 – 269 - Please describe the water condition parameters during experiment.
8. Lines 291-294 – Please add information about laboratory equipment used during genetic analyses (PCR, sequencing/genotyping).
Author Response
Dear Reviewer,
First of all, we are very grateful for your recognition of our research work. Thank you very much for your comments and suggestions. The valuable comments from you not only helped us with the improvement of our manuscript, but suggested some ideas for future studies. We added a section about the limitations and prospects in the discussion of this article to address your comments.
Below you will find our responses to your comments:
- Line 36 - please explain if there are different times/seasons of breeding prawns. If so, please include it in the text. Please also include information on the seasonality of reproduction, including the length and body weight of organisms, and indicating the water temperature.
Response: Thanks for your kindly reminding. We add a paragraph to address your comments in the introduction in revised manuscript.
- nipponense has the characteristic of rapid sexual maturation. The breeding season of M. nipponense is from April to October every year (with the water temperature at 22 to 30 ℃). The overwintering adult female prawns (BW±SD: 0.78±0.23 g, BL±SD: 28.58±3.35 cm) enters into the breeding season at the end of April and its sexual maturation and embryo development will be accelerate by the rising of the water temperature. Every year, the newborn prawns in July in the pond can reproduce a large number of offspring in autumn (August to September, water temperature at 28 to 30 ℃, BW±SD: 0.50±0.15 g, BL±SD: 25.90±2.12 cm) (commonly known as “autumn reproduction”).
- Paragraph in lines: 44-54 - Please describe clear differences and similarities between crustaceans and fish in terms of reaching sexual maturity at a given age in relation to body size. Moreover, the sentence (lines: 48-48) is unclear. Please explain the height of reproductively mature individuals.
Response: Thanks for your good comments. We rewrote this paragraph to clearly describe the differences and similarities between crustaceans and fish in terms of reaching sexual maturity at a given age in relation to body size and made the reference presentation more clear. Please find the revised paragraph as followed.
The gonad development is the basis for the reproductive behavior of fishery resources and it is closely related to body size and age of individuals. In fish, most studies showed that the first sexual maturation is related to body size (body length). Primary maturation begins when the size of the fish reaches or exceeds the minimum body length required for sexual maturity, and then the proportion of mature individual’s increases with the increase of body length. There were a few researches indicated that the first sexual maturity of fish, such as Maccullochella peeli peeli and Larimichthys crocea, were closely related to age [5-7]. In L. crocea, it had been proved to be negative correlation with the body weight and selective breeding for the specific gonadosomatic index (GSI) is beneficial to improve the dressing percentage [6, 7]. In crustaceans, there were a little reports about relationship between the first sexual maturity and body size and age, confirming that body size is more vital to the age in Penaeus schmitti, Oratosquilla oratoria and Trachysalambria curvirostris [8-10]. In T. curvirostris, the growth of individuals were slowed but not stop completely during the sexual maturity [10]. In M. nipponense, the GSI had a significant negatively effect on the female prawn size [11]. Therefore, elucidating the correlation of gonadal development and growth of fishery resources could provide a scientific basis for resource assessment and breeding management.
- Paragraph in lines: 44-54 – please add the fragment of text about GSI in crustacean. This is very important information to refer to the data regarding the large yellow croaker.
Response: Thanks for your good comments. We add a reference about GSI in crustacean. Please find the revised paragraph in question 2.
- Lines 58-60 – Please add some explanations of various mechanisms involved in growth of M. nipponense.
Response: Thanks for your kindly reminding. We revised this sentence to “Various factors, such as temperature, dietary and salinity, involved in the growth of M. nipponense have been reported [13-17].”
- Line 263 – Please add the information about age and length of selected M. nipponense.
Response: Thanks for your comments. We added the body length and age.
Healthy adult M. nipponense [n=50, body weight (BW ± SD: 1.32 ± 0.17 g; BL ± SD: 3 0.24 ±3.20 g, 10 month age) were selected from the Freshwater Fisheries Research Center Dapu Scientific Experimental Base (Wuxi, China).
- Paragraph 4.3. - The information about the method of measurements of: total length (TL), BW, body length (BL), second pereiopod length (SPL), abdomen width (AW), abdomen length (AL), carapace width should be included (CW), and carapace length (CL) in this paper.
Response: Thanks for your comments. We added the information about the method of measurements as followed.
Body weight (BW): the samples were weighted by removing water. Total length (TL): from the rostral front end to end of the telson. Body length (BL): from the boundary of the post-eyestalk to the tip of the telson. Second pereiopod length (SPL): from the second pereiopod base to its end. Abdomen width (AW): the width of the third abdomen. Abdomen length (AL): from the first abdomen front to the sixth abdominal segment end. Carapace width (CW): the width of carapace at the fifth pereiopod. Carapace length (CL): from the tip of the rostral front to the central end of the back of the carapace.
- Lines 266 – 269 - Please describe the water condition parameters during experiment.
Response: Thanks for your kindly reminding. We added the water condition parameters during experiment in the revised manuscript.
The Water temperature from 22 to 31 ℃, the pH was from 6.7 to 7.5, the dissolved oxygen content was from 4.5 to 11.7 mg/L, and the ammonia nitrogen content was from 0.25 to 0.81 mg/L.
- Lines 291-294 – Please add information about laboratory equipment used during genetic analyses (PCR, sequencing/genotyping).
Response: Thanks for your kindly reminding. We added the information about laboratory equipment used during genetic analyses in the revised manuscript.
The PCR cycling conditions were 94℃ for 4 min, followed by 30 cycles of 94℃ for 5 s, 60 ℃ for 30s and 72℃ for 30s, then 72℃ 10min (Eppendorf, German). The PCR products were tested by 1.2% agarose gel and then send to Sangon Biotech (Shanghai, China) for sequencing. The sequencing results were compared by MEGA 11.0 software to screen SNP sites.
Reviewer 2 Report
Comments and Suggestions for Authors
In this study, the association between Mn-CTS L1 polymorphism and gonad development and growth traits of M. nipponense was investigated. Some valuable SNPs were obtained, which could be applied as the potential molecular markers for gene-assisted selection for M. nipponense. There are still some issues that need to be fixed in the manuscript.
1. Table 1, there are two Total length and two Body length in the table. What do they represent?
2. Please add units to the values in Table 1, 2 and 4.
3. Table 4 and 5, there are some values for some genotype not grouped. Do they mean that there is no significant difference of the genotypes with the others for one same SNP site.
4. Materials and Methods, after the PCR amplification, how do you obtain the SNPs? By sequencing? If so, please provide sequencing platform information and method for SNP calling.
Author Response
Dear Reviewer,
First of all, we are very grateful for your recognition of our research work. Thank you very much for your comments and suggestions. The valuable comments from you not only helped us with the improvement of our manuscript, but suggested some ideas for future studies. We added a section about the limitations and prospects in the discussion of this article to address your comments.
Below you will find our responses to your comments:
- Table 1, there are two Total length and two Body length in the table. What do they represent?
Response: Thanks for your kindly reminding. We corrected this mistake in the revised manuscript.
- Please add units to the values in Table 1, 2 and 4.
Response: Thanks for your kindly reminding. We added units to the values in Table 1, 2 and 4.
- Table 4 and 5, there are some values for some genotype not grouped. Do they mean that there is no significant difference of the genotypes with the others for one same SNP site.
Response: Thanks for your kindly reminding. We corrected the significant difference description in Table 4 and 5.
- Materials and Methods, after the PCR amplification, how do you obtain the SNPs? By sequencing? If so, please provide sequencing platform information and method for SNP calling.
Response: Thanks for your kindly reminding. We added the information about laboratory equipment used during genetic analyses in the revised manuscript.
The PCR cycling conditions were 94℃ for 4 min, followed by 30 cycles of 94℃ for 5 s, 60 ℃ for 30s and 72℃ for 30s, then 72℃ 10min (Eppendorf, German). The PCR products were tested by 1.2% agarose gel and then send to Sangon Biotech (Shanghai, China) for sequencing. The sequencing results were compared by MEGA 11.0 software to screen SNP sites.